# Epidemiology of Shiga Toxin-Producing *Escherichia coli* O157 in the Province of Alberta, Canada, 2009–2016

**DOI:** 10.3390/toxins11100613

**Published:** 2019-10-22

**Authors:** Luiz F. Lisboa, Jonas Szelewicki, Alex Lin, Sarah Latonas, Vincent Li, Shuai Zhi, Brendon D. Parsons, Byron Berenger, Sumana Fathima, Linda Chui

**Affiliations:** 1Department of Laboratory Medicine and Pathology, Faculty of Medicine and Dentistry, University of Alberta, Edmonton, AB T6G 2R3, Canada; jszelewi@ualberta.ca (J.S.); alex.lin@albertapubliclabs.ca (A.L.); szhi@ualberta.ca (S.Z.); brendonp@dal.ca (B.D.P.); byron.berenger@albertapubliclabs.ca (B.B.); 2Faculty of Medicine and Dentistry, University of Alberta, Edmonton, AB T6G 2R3, Canada; sarah.latonas@cls.ab.ca; 3Provincial Laboratory for Public Health, Alberta Public Laboratories, Edmonton, AB T6G 2B7, Canada; vincent.li@albertapubliclabs.ca; 4The Ministry of Health, Government of Alberta, Edmonton, AB T5J 1S6, Canada; sumana.fathima@gov.ab.ca

**Keywords:** enterohemorrhagic *Escherichia coli*, Shiga toxins, virulence factors, human infection

## Abstract

Shiga toxin-producing *Escherichia coli* (STEC) infections are the product of the interaction between bacteria, phages, animals, humans, and the environment. In the late 1980s, Alberta had one of the highest incidences of STEC infections in North America. Herein, we revisit and contextualize the epidemiology of STEC O157 human infections in Alberta for the period 2009–2016. STEC O157 infections were concentrated in large urban centers, but also in rural areas with high cattle density. Hospitalization was often required when the Shiga toxin genotype *stx*2a *stx*2c was involved, however, only those aged 60 years or older and infection during spring months (April to June) independently predicted that need. Since the late 1980s, the rate of STEC O157-associated hemolytic uremic syndrome (HUS) in Alberta has remained unchanged at 5.1%, despite a marked drop in the overall incidence of the infection. While Shiga toxin genotypes *stx*1a *stx*2c and *stx*2a *stx*2c seemed associated with HUS, only those aged under 10 years and infection during spring months were independently predictive of that complication. The complexity of the current epidemiology of STEC O157 in Alberta highlights the need for a One Health approach for further progress to be made in mitigating STEC morbidity.

## 1. Introduction

Shiga toxin-producing *Escherichia coli* (STEC) is a major cause of foodborne disease in humans and is associated with morbidity and mortality worldwide [1]. While meats are frequent sources of infection, contaminated water and produce from livestock-producing areas are also implicated in human disease. The burden of STEC on the health of Canadians and on the healthcare system is considerable [2,3]. This may be especially true in jurisdictions directly involved in the production of food implicated in STEC transmission.

The province of Alberta continues to be the leading beef producer in Canada [4]. Shortly after detection of the first few cases of STEC in the country, Alberta established an enhanced surveillance system based on case detection and reporting by both healthcare providers and microbiology laboratories, which in turn allowed for the description of the epidemiology of STEC infections in the late 1980s in one of the first STEC population-based studies [5]. Public health microbiology has dramatically changed since then. PCR-based detection of Shiga toxins 1 (*stx*1) and 2 (*stx*2) genes has been made available as well as internationally standardized protocols for molecular strain typing by pulsed-field gel electrophoresis (PFGE) and multilocus variable-number tandem repeat analysis (MLVA). Shiga toxin genes can now be assigned to three *stx*1 (1a, 1c, 1d) and seven *stx*2 (2a, 2b, 2c, 2d, 2e, 2f, 2g) subtypes [6], thus allowing for further molecular characterization of these important virulence determinants and understanding of the epidemiology of these STEC isolates. The availability of STEC isolates from our surveillance program creates a unique opportunity to study the associations between isolate relatedness, genotype, and associated morbidity in our province.

With the objectives of revisiting the epidemiology of *E. coli* O157 infection in Alberta and assessing for patient and bacterial characteristics predictive of infection-associated morbidity, we retrospectively reviewed the STEC O157 cases in the province of Alberta over an eight year period, between the years 2009 and 2016.

## 2. Results

A review of the public health database for cases of STEC O157 infections in Alberta for the period 2009 to 2016 yielded a total of 801 cases. Overall, the annual incidence of STEC O157 infections remained stable at around two cases/100,000 persons per year (range 1.71–2.61/100,000), with the exception of 2014, when the third largest STEC outbreak in Canadian history led to the highest incidence seen during the study period (4.6/100,000) [7]. All viable STEC O157 isolates available on culture-confirmed cases (n = 762) were included in the study.

### 2.1. STEC O157 Isolate Characteristics

Serotyping of the flagellar antigen H revealed the predominance of the serotype H7 (97.9%), with rare instances of non-motile (1.8%) and motile isolates for which the H antigen was untypeable (0.3%). Characterization of virulence genes by PCR demonstrated the presence of the locus of enterocyte effacement gene intimin (*eae*) in all isolates. *stx*1 was present in 76.2% of the isolates, whereas *stx*2 was present in 100% of the isolates. *stx*1a *stx*2a was the predominant Shiga toxin genotype, seen in 72.2% of the isolates, followed by *stx*2a in 13.4%, *stx*2a *stx*2c in 6.6%, *stx*2c in 3.9%, *stx*1a *stx*2c in 2.2%, and *stx*1a *stx*2a *stx*2c in 1.6%.

Isolate relatedness was characterized by PFGE and MLVA patterns, and further supported by temporal and epidemiological linkage to additional case(s). Evidence of epidemiological linkage was available for 228/762 cases involved in one of the 27 provincial, national, or international (USA) outbreaks investigated during the 2009–2016 period. As expected, due to variable frequency and size of the outbreaks, the proportion of outbreak-related and sporadic isolates varied from year to year. Overall, sporadic isolates (45.8%) were nearly as frequent as isolates genetically, and temporally/epidemiologically related (i.e., outbreak-related isolates, 54.2%).

### 2.2. Outcomes of STEC O157 Infection and Demographic Characteristics of Cases

STEC O157 infections similarly affected male and female sex (M:F ratio 1:1.23) and disproportionately affected those aged 19 years or younger, who represented nearly half (47.2%) of the cases. Hospitalization due to STEC O157 infection was required in 25.1% of the cases, and was most frequent among children under the age of 10 years (n = 69/101, 36.1%) and among adults aged 60 years or older (n = 41/101, 21.5%). Hemolytic uremic syndrome (HUS) occurred as a complication in 5.1% of the cases (n = 39). Nearly 80% percent of the cases of HUS (n = 31/39, 79.4%) were observed in children under the age of 10 years. Over half (53.8%) of the HUS cases were seen in patients infected with outbreak-related, with the remaining HUS cases associated with sporadic STEC O157 isolates.

### 2.3. Seasonality of STEC O157 Infection and Its Geographic Distribution

Infection activity was documented throughout the year, peaking during the summer months (July–September) (n = 428/762, 56.2%), coinciding with the peak of STEC O157-related hospitalization (n = 101/191, 52.9%) and with the peak of HUS cases (n = 21/39, 53.8%). The remaining cases of infection were nearly evenly distributed between the remaining seasons (118/762, 15.5% in October–December; 90/762, 11.8% in January–March; 126/762, 16.5% in April–June). The remaining cases requiring hospitalization were also nearly evenly distributed between the remaining seasons (29/191, 15.2% in October–December; 20/101, 10.5% in January–March; 41/101, 21.5% in April–June). Despite 118 cases of infection recorded in the fall months (October–December), only one out of 39 cases of HUS was observed, with the remaining HUS cases distributed between January–March (n = 5/39, 12.8%) and April–June (n = 12/39, 30.8%).

STEC O157 infections were concentrated in the health administrative zones encompassing the two largest urban centers in the province: Calgary (42.4%) and Edmonton (23.5%). The remainder of the cases was distributed across the predominantly rural North (4.7%), Central (10.2%), and South (19.2%) zones. The incidence of STEC infection during the period increased in a north-to-south fashion, with a median of 0.99 cases/100,000 persons per year in the North (range 0–1.84/100,000); 1.47/100,000 in Edmonton (range 0.52–4.53/100,000); 1.90/100,000 in the Central (range 1.34–3.58/100,000); 2.25/100,000 in Calgary (range 1.51–5.53/100,000); and 6.60/100,000 in the South (range 4.15–7.27/100,000) zones. This pattern was approximately mirrored by regional cattle-to-human density, as illustrated in Figure 1 for 2016.

Hospitalization and HUS were both disproportionally concentrated in the South (22% of hospitalizations, 28.2% of HUS) and Central (15.2% of hospitalizations, 17.9% of HUS) zones, considering that their population is approximately 3–5 times smaller than that from Calgary (34.6% of hospitalizations, 28.2% of HUS) and Edmonton (20.4% of hospitalizations, 25.6% of HUS) zones. No cases of HUS were observed in the North zone, which responded to 7.9% of hospitalizations.

### 2.4. Predictors of STEC O157 Morbidity

The morbidity associated with STEC O157 infections was further investigated for putative associations with bacterial and infection characteristics, after the exclusion of patients for whom the outcomes of hospitalization and HUS were unknown (n = 38, 5.0%).

Predictors of HUS were investigated in uni- and multivariate analysis (Table 1). Patients under the age of five (OR 9.311, CI 95% 3.588–24.161, *p* < 0.001) or between the ages of five and nine (OR 5.208, CI 95% 1.791–15.143, *p* = 0.002) had a significantly higher risk of HUS. A trend of association between infection diagnosed in the months of April to June and HUS was also seen (OR 2.314, CI 95% 1.066–5.024, *p* = 0.034). Shiga toxin genotypes, although statistically associated with the occurrence of HUS in univariate analysis, did not independently predict that outcome. Sex and infection with an outbreak-related isolate showed no association with HUS. 

Similarly, predictors of hospitalization were also analyzed (Table 2). Patients aged 60 years or older (OR 3.054, CI 95% 1.879–4.961, *p* < 0.001), or diagnosed between the months of April to June (OR 1.795, CI 95% 1.157–2.786, *p* = 0.009) had a significantly higher need of hospitalization. A trend toward a lower need of hospitalization was seen for patients aged 40 to 49 years (OR 0.328, CI 95% 0.114–0.943, *p* = 0.039), and a trend toward higher need of hospitalization was seen for infections involving isolates of genotype *stx*2a *stx*2c (OR 1.878, CI 95% 1.006–3.505, *p* = 0.048). Infection with outbreak-related STEC O157 isolates, although statistically associated with the need of hospitalization in univariate analysis, was not predictive of that outcome. Sex was not associated with the need for hospitalization.

## 3. Discussion

In the late 1980s to early 1990s, the province of Alberta documented one of the highest rates of STEC infections in North America. Local incidence of STEC O157 is now much lower, largely due to improved laboratory detection of cases leading to public health interventions targeting the prevention, detection, and remediation of outbreaks [7]. Herein we report on the updated local epidemiology of STEC O157 in the context of molecular data available on a large collection of clinical isolates from our Alberta STEC surveillance program.

STEC is associated with multiple disease states, the most clinically significant being HUS. Interestingly, despite a marked reduction in the incidence of STEC O157 infections, HUS rates have remained unchanged in comparison with an earlier report [5], and continue to affect approximately 5% of those infected. Antibiotic use in STEC infections is a well-known risk factor for the development of HUS, and in the absence of antibiotic usage data in the present work, we can only hypothesize that it may contribute to the stagnant rates of HUS [8,9], as 25% of children with STEC infections in the province of Alberta still receive antibiotics including 50% of those developing HUS [10]. Healthcare provider education and their adherence to practice guidelines would be necessary in order to reduce such inappropriate usage of antibiotics. Future iterations of STEC epidemiology work, in Alberta or elsewhere, would greatly benefit from the linkage between antibiotic consumption, clinical outcomes, and molecular data on bacterial isolates. This would allow for the simultaneous assessment of the relative contribution of virulence factors and antibiotic use in the development of HUS across different age groups. Another factor potentially hindering further reductions in HUS rates is delayed diagnosis, which has traditionally relied on culture-based organism isolation and/or Shiga toxin detection, as reflected in the period (2009–2016) studied. An opportunity for timely diagnosis lies on the frontline use of PCR-based multiplex panels for the detection of enteric pathogens, coupled with the reflex culture of PCR-positive specimens. Early presumptive molecular diagnosis within hours of patient presentation, instead of within days as required for culture-based diagnosis, may improve antimicrobial stewardship in STEC infections, while allowing for earlier interventions such as fluid replacement capable of reducing the morbidity associated with STEC HUS [11], but without negatively interfering with public health surveillance [12].

Timely diagnosis and appropriate clinical management may not prevent all cases of HUS, evidencing the relevance of the interplay between the pathogen and host factors leading to that outcome. The pathogen-attributable risk of HUS is determined by the overall strain virulence, which in turn is the product of the combination and expression levels of multiple bacterial and phage virulence gene products. Shiga toxin is the most readily recognized phage-derived virulence factor implicated in the pathophysiology of HUS, yet many other contributing factors are also relevant to the development of this complication [13]. Since HUS risk-assessment based on multiple virulence markers remains impractical and unaffordable to perform in routine patient care, PCR or antigen-based detection of Stx1 and Stx2 can be used as a surrogate marker, confirming infection with Shiga toxin-producing strains. Further differentiation of Shiga toxins into specific subtypes, however, may be clinically and epidemiologically informative.

For instance, our data suggested a statistical association between the *stx*1a *stx*2c or the *stx*2a *stx*2c Shiga toxin genotypes and increased odds of HUS. Isolates encoding *stx*2a, the Shiga toxin subtype with the highest cytotoxic potential [14] and classically associated with the risk of HUS, were involved in approximately 85% of HUS cases. *stx*2c, although of comparatively lower cytotoxicity and higher heat lability [15], was involved in over 30% of cases of HUS, not all of them justifiable by the concomitant presence of other Shiga toxin subtypes. No cases of HUS were documented among patients infected with the *stx*1a *stx*2a *stx*2c genotype, despite the combination of three different toxins, and those infected with the *stx*1a *stx*2a genotype isolates were statistically less likely to develop HUS, regardless of the presence of *stx*2a. The cytotoxicity potential of individual Shiga toxin subtypes or the sum of their effects when combined are clearly insufficient to explain the morbidity resulting from STEC O157 infections. Shiga toxin expression levels and virulence factors encoded by the *E. coli* O157 strains must therefore be involved.

Shiga toxins are encoded by lambda bacteriophages, and their expression levels seem regulated, at least in part, by the bacteriophage-encoded anti-terminator *q* gene. Higher levels of *stx*2a expression in STEC O157 have been associated with the anti-terminator *q*933 [16,17], whereas lower levels of *stx*2a expression have been associated with the presence of the anti-terminator *q*21 and the presence of mutations on the *stx*2a promoter [18]. Expression of *stx*2c has been reported as comparatively lower than *stx*2a [19], but the role of the *stx*2c-associated anti-terminator *q*2851 has been less well characterized [20,21].

Associations between genetic lineage, a determinant of bacterial strain virulence, Shiga toxin genotype, and disease severity have been previously reported. Lineage and Shiga toxin genotype may be closely connected as described in the state of Washington, USA, where *stx*1a *stx*2a was associated with the sublineage Ib (various clades) in 90.1%, *stx*2a *stx*2c with sublineage IIa (clade 8) in 83.7%, and *stx*2a with sublineage IIb (clade 8) in 94.4% of cases, with *stx*1a *stx*2c and *stx*2c genotypes associated with rare lineages corresponding to clade 7 [22]. Previous work characterizing both the lineages and *stx* subtypes of Canadian human and cattle STEC O157 including isolates from Alberta, suggested the presence of *stx*2c in nearly all (96.7%) isolates belonging to lineage-specific polymorphism assay (LSPA6) lineage II, in approximately 50% of isolates belonging to lineage I/II and only rarely (1.8%) in isolates of lineage I [23]. It is thus plausible that isolates from the present study encoding *stx*2c may belong to lineages II or I/II. Lineage I/II [24], and sublineages IIa and IIb [22], corresponding to clade 8, have been implicated in severe disease including HUS [25,26], with severity of disease modulated by association with specific Shiga toxin subtypes [27]. Our study was limited in that the population structure of the STEC O157 isolates was not further characterized, therefore not allowing for direct inferences on clone-associated virulence on this dataset.

Our results also point toward an important host-related risk factor for STEC O157-associated morbidity: age. Children under the age of 10 years had a significantly increased risk of HUS, whereas adults aged 60 years or older had a significantly higher need of admission to hospital during the episode of STEC O157 infection. Young [28] and old age [29] have been long recognized for their association with STEC-related morbidity and mortality. While recognized as a risk factor on its own, we now learn that age may actually interact with other risk factors in determining the overall risk of STEC O157 morbidity. For example, a stronger association between clade 8 isolates and HUS was seen among children under the age of 10 years than among those aged 10 years or older [26]. Additionally, age appeared to be an effect-modulating factor for the association between genetic lineage and the risk of HUS [22]. With the ongoing transition of molecular surveillance from PFGE and MLVA to whole genome sequencing for STEC surveillance in Canada, future iterations of epidemiological studies of STEC infections should be able to address the gap in knowledge about genetic lineages involved in STEC morbidity in our country.

Finally, an appreciation of the connections between humans, animals, and the environment is warranted. In Alberta, the majority of STEC O157 isolated from humans share the same lineage [30] and same Shiga toxin genotype [31] with isolates from cattle, corroborating the latter as an important direct and/or indirect source of infection to the former. This complex interaction is modulated by climate and weather, with STEC O157 genetic diversity in cattle varying over the course of the seasons [32], and with the intensity of bacterial shedding modified by severe weather [33]. These dynamics may help explain our findings of an independent association between infection during spring months and the significantly increased likelihood of hospitalization and a trend toward the increased likelihood of HUS, as may also, perhaps, the unexpectedly lower rates of HUS during the fall months. Indeed, the seasonal occurrence of HUS has been previously described [5,34,35]. However interested in assessing the interactions between the four seasons and the six isolated Shiga toxin genotypes documented in the present data, such analysis was limited by the overall (72.2% of all isolates) and seasonal (67.8 to 78.6% of the isolates in a given season) predominance of *stx*1a *stx*2a isolates, and the small numbers of isolates of each remaining genotype in a given season. Larger datasets are required to adequately investigate associations between bacterial and Shiga toxin genotypes, and the seasonality of STEC infections.

Cattle density measurements themselves have also been previously shown to correlate with STEC O157 infection rates in humans [36,37]. Not surprisingly, STEC O157 infections in Alberta were not restricted to high human density large urban centers, but also observed in rural areas where, relative to human presence, cattle density is elevated. In association with cattle farming, contamination of the watershed in Alberta’s southernmost region has been documented [38] to affect irrigation water [39], thereby also extending the STEC O157 contamination potential to produce and recreational waters. Environmental contamination may provide an alternative path for infection other than the ingestion of contaminated meat. Efforts targeting animal health and the environment are clearly required to limit the STEC O157 burden on human health.

Overall, we provided an updated epidemiological picture of STEC O157 infections in Alberta with the contribution of molecular techniques, several years after the first population-based study describing this issue in our province. Although local infection rates have largely decreased since the emergence of STEC O157 as an issue of major public health importance, much has yet to be done in order to further reduce the stagnant rates of HUS. While the clinical microbiology laboratory landscape is quickly evolving in response to this demand, a One Health approach is critical for the successful control of STEC O157 infections.

## 4. Materials and Methods

### 4.1. Setting

Infection by Shiga toxin-producing *E. coli* strains is a notifiable disease in the province of Alberta, Canada, and is reportable to the local health authority. Throughout the study period (2009–2016), the network of Alberta sentinel microbiology laboratories adhered to the Centers for Disease Control and prevention (CDC) recommendations for the isolation of STEC [40] through the use of selective differential agar (sorbitol-MacConkey agar or CHROMagar O157) for the primary plating of all stool samples submitted for culture. All *E. coli* O157 isolates were referred to the Provincial Laboratory for Public Health (ProvLab) for confirmation and further characterization to support surveillance and monitoring.

### 4.2. Data Collection

Procedures were reviewed and approved by the University of Alberta Human Research Ethics Board. STEC isolates received at ProvLab for confirmatory testing and organism typing were prospectively catalogued upon receipt, along with respective test results. Demographic and hospitalization data temporally-associated with the STEC infection episode and/or associated HUS are prospectively captured by The Ministry of Health as per the notifiable disease reporting regulations [41]. Mortality attributable to STEC infections and renal replacement therapy requirements secondary to HUS are not recorded. After deterministic linkage of microbiological and public health data, all individual-level data were anonymized prior to use in this study.

### 4.3. Case Definitions and Inclusion Criteria

Provincial STEC infection case definitions included both confirmed (laboratory confirmation of infection, with or without clinical illness by means of isolation of STEC in culture, and/or detection of Shiga toxin antigen or nucleic acid in an appropriate clinical specimen) and probable (compatible clinical illness in a person epidemiologically linked to a confirmed case) cases. STEC O157 infections diagnosed in Alberta residents by means of a positive stool culture collected between January 2009 and December 2016 were included. Given the aim of retrospective characterization of Shiga toxin subtypes in clinical specimens, only confirmed STEC infections for which viable bacterial isolates were available for additional testing were included. Cases in which ProvLab was unable to grow a previously cryopreserved isolate were excluded. Further, given regional and temporal differences in microbiological testing approaches impacting the detection of non-O157 STEC isolates (i.e., use of sorbitol-MacConkey agar or CHROMagar O157), only STEC O157 infections were analyzed, and non-O157 infections were excluded. As a consequence, only confirmed cases of hemolytic uremic syndrome (HUS) associated with STEC O157 were retained in the dataset, defined by the acute onset of microangiopathic hemolytic anemia, thrombocytopenia, and acute renal impairment starting within three weeks after the onset of a diarrheal or non-diarrheal illness and occurring in the absence of chronic underlying conditions that may account for renal and hematological dysfunction.

### 4.4. Virulence Gene Detection and Subtyping

*E. coli* O157 isolates were retrieved from frozen stock cultures previously stored at −80 °C and subcultured twice onto 5% sheep blood agar. Single-colony DNA extraction using Triton X-100 rapid lysis buffer was performed as previously described [42], and the supernatant obtained was utilized as the template for all subsequent PCR assays. Primers and probes utilized are summarized in Appendix A. The presence of the locus of enterocyte effacement gene *eae* was confirmed by qPCR as described by Nielsen and Andersen [43]. The presence of the Shiga toxin 1 and 2 genes (*stx*1 and *stx*2) were first confirmed on isolates from 2009 to 2016 by qPCR as previously described [44] and followed by the subtyping scheme described by Scheutz et al. [6]. Based on previously published Shiga toxin subtyping data from Alberta on STEC, only subtypes 1a, 2a, and 2c were observed [45]; therefore, a simplified subtyping strategy was devised, targeting *stx* 1a, 2a, and 2c genotyping. For isolates from 2009 to 2015, s*tx*1a and *stx*2a subtyping was performed by conventional PCR [6]. For isolates from 2016, s*tx*1a, *stx*2a subtyping was performed by qPCR using locked nucleic acid (LNA) probes [46]. *Stx*2c subtyping was performed for all isolates by SYBRGREEN qPCR using primers previously described by Zhi et al. [46].

### 4.5. Bacterial Typing and Clustering Analysis

Bacterial typing by pulsed-field gel electrophoresis for *E. coli* O157 was locally performed using standardized PulseNet International protocols [47]. Fingerprinting profiles generated using *Xba*I/*Bln*I endonuclease restriction digests were analyzed using BioNumerics v6.01 software (Applied Maths, Austin, TX, USA) and national fingerprinting patterns were assigned. Further typing data by multilocus variable number tandem repeats (MLVA) analysis was performed by the PulseNet Canada Laboratory using the PulseNet standardized protocol [48].

Isolates were classified as outbreak-related or sporadic, on the basis of shared PFGE and MLVA molecular signatures and/or epidemiological link, and co-occurrence within a predetermined timeframe. MLVA profiles were utilized to increase the resolution of *Xba*I PFGE clusters as follows. Isolates with indistinguishable PFGE *Xba*I pattern and MLVA (i), indistinguishable PFGE *Xba*I pattern plus highly similar MLVA (i.e., the same number of repeats in VNTR locus 34 and maximum one repeat difference in up to three VNTR loci or up to three repeats difference in a single VNTR locus) (ii), or closely related PFGE *Xba*I (i.e., up to two band difference) and identical MLVA (iii) were considered to be outbreak-related when specimens were collected within a 60-day period, or within 60 days from the collection date of the last cluster isolate. Isolates fulfilling none of the three molecular criteria above but having a documented epidemiological link with a cluster of cases were also deemed to be outbreak-related. All remaining isolates were deemed to be sporadic in their occurrence.

### 4.6. Geographical Analysis

Spatial distribution of cattle in the province was characterized using the geographical information system QGIS 3.0 [49]. Shapefiles were obtained from the provincial Office of Statistics & Information [50]. Cattle data were obtained via provincial open government program [51]. Province population estimates were obtained from the provincial Interactive Health Data Application [52].

### 4.7. Statistical Analysis

Statistical analyzes were performed on IBM SPSS Statistics 25 (SPSS Inc., Chicago, IL, USA). Categorical variables were compared using the chi-square or Fisher-Exact tests. Multivariate logistical regression models were performed using the entry method, with the inclusion of only variables with *p* < 0.05 in univariate analysis, with two-tailed *p* < 0.01 considered to be significant.

## Figures and Tables

**Figure 1 toxins-11-00613-f001:**
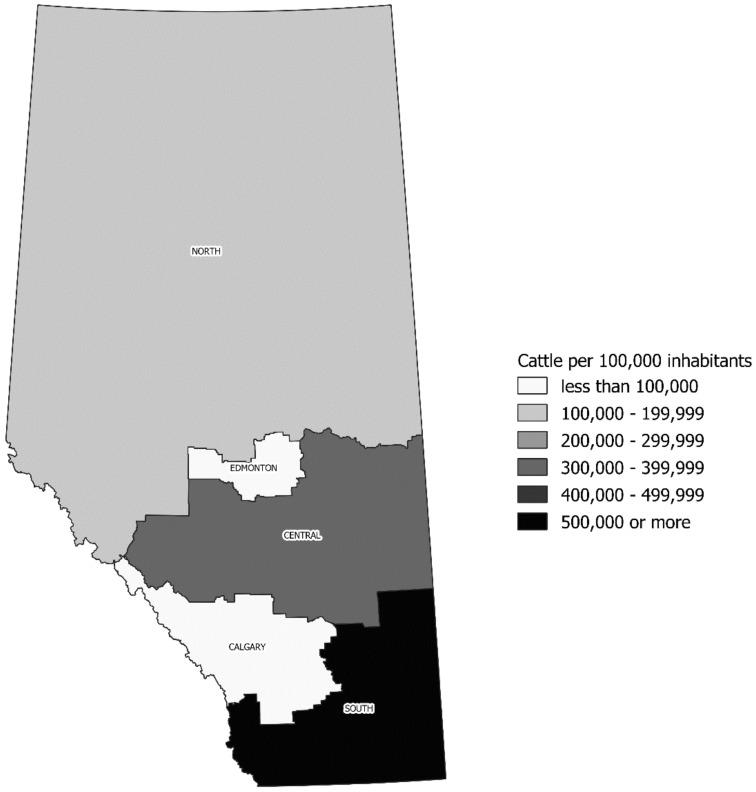
Density of cattle in Alberta health administrative areas per 100,000 inhabitants for 2016.

**Table 1 toxins-11-00613-t001:** Univariate and multivariate analysis of predictors of hemolytic uremic syndrome (HUS).

Variable	With HUS	Without HUS	Univariate OR (CI 95%)	*p*-Value	Multivariate OR (CI 95%)	*p*-Value
Sex				0.793		
Female	21 (5.2)	383 (94.8)	0.917 (0.480–1.752)			
Male	18 (5.6)	301 (94.4)	1.091 (0.571–2.084)			
Age group						
<5	21 (14.6)	123 (85.4)	5.319 (2.75–10.309)	<0.001	9.311 (3.588–24.161)	<0.001
5–9	10 (10.3)	87 (89.7)	2.364 (1.11–5.025)	0.021	5.208 (1.791–15.143)	0.002
10–19	2 (2.1)	95 (97.9)	0.335 (0.079–1.414)	0.148		
20–29	2 (1.3)	155 (98.7)	0.184 (0.044–0.774)	0.008	0.776 (0.153–3.942)	0.76
30–39	0	50 (100.0)		0.102		
40–49	0	40 (100.0)		0.159		
50–59	2 (3.5)	55 (96.5)	0.618 (0.145–2.632)	0.511		
≥60	2 (2.5)	79 (97.5)	0.414 (0.098–1.751)	0.299		
Season						
Jan–Mar	5 (5.9)	80 (94.1)	1.110 (0.422–2.924)	0.798		
Apr–Jun	12 (10.2)	106 (89.8)	2.421 (1.190–4.926)	0.012	2.314 (1.066–5.024)	0.034
Jul–Sep	21 (5.2)	385 (94.8)	0.906 (0.474–1.730)	0.765		
Oct–Dec	1 (0.9)	113 (99.1)	0.133 (0.018–0.978)	0.02	0.191 (0.025–1.454)	0.11
Shiga toxin genotype						
*stx*1a *stx*2a	20 (3.8)	504 (96.2)	0.376 (0.196–0.720)	0.002	0.466 (0.198–1.097)	0.08
*stx*1a *stx*2a *stx*2c	0	12 (100)		1		
*stx*1a *stx*2c	4 (25.0)	12 (75.0)	6.410 (1.965–20.833)	<0.001	2.679 (0.636–11.277)	0.179
*stx*2a	7 (7.4)	88 (92.6)	1.481 (0.635–3.460)	0.361		
*stx*2a *stx*2c	6 (12.5)	42 (87.5)	2.778 (1.103–6.993)	0.024	1.657 (0.502–5.465)	0.407
*stx*2c	2 (7.1)	26 (92.9)	1.368 (0.313–5.988)	0.659		
Isolate relatedness				0.933		
Outbreak-related	21 (5.3)	373 (94.7)	0.973 (0.509–1.858)			
Sporadic	18 (5.5)	311 (94.5)	1.028 (0.538–1.965)			

**Table 2 toxins-11-00613-t002:** Univariate and multivariate analysis of predictors of hospitalization.

Variable	Hospitalized	Not Hospitalized	Univariate OR (CI 95%)	*p*-Value	Multivariate OR (CI 95%)	*p*-Value
Sex				0.217		
Female	114 (28.2)	290 (71.8)	1.235 (0.883–1.728)			
Male	77 (24.1)	242 (75.9)	0.809 (0.579–1.132)			
Age group						
<5	39 (27.1)	105 (72.9)	1.044 (0.692–1.575)	0.84		
5–9	30 (30.9)	67 (69.1)	1.294 (0.811–2.062)	0.279		
10–19	19 (19.6)	78 (80.4)	0.643 (0.378–1.094)	0.101		
20–29	32 (20.4)	125 (79.6)	0.655 (0.427–1.007)	0.053		
30–39	7 (14.0)	43 (86.0)	0.433 (0.191–0.979)	0.039	0.445 (0.194–1.021)	0.056
40–49	4 (10.0)	36 (90.0)	0.295 (0.103–0.840)	0.015	0.328 (0.114–0.943)	0.039
50–59	19 (33.3)	38 (66.7)	1.437 (0.806–2.558)	0.217		
≥60	41 (50.6)	40 (49.4)	3.367 (2.096–5.405)	<0.001	3.054 (1.879–4.961)	<0.001
Season						
Jan–Mar	20 (23.5)	65 (76.5)	0.840 (0.494–1.429)	0.52		
Apr–Jun	41 (34.7)	77 (65.3)	1.616 (1.059–2.463)	0.025	1.795 (1.157–2.786)	0.009
Jul–Sep	101 (24.9)	305 (75.1)	0.835 (0.599–1.164)	0.288		
Oct–Dec	29 (25.4)	85 (74.6)	0.942 (0.595–1.488)	0.796		
Shiga toxin genotype						
*stx*1a *stx*2a	140 (26.7)	384 (73.3)	1.058 (0.729–1.536)	0.767		
*stx*1a *stx*2a *stx*2c	1 (8.3)	11 (91.7)	0.249 (0.032–1.946)	0.199		
*stx*1a *stx*2c	5 (31.3)	11 (68.8)	1.274 (0.437–3.717)	0.774		
*stx*2a	21 (22.1)	74 (77.9)	0.765 (0.457–1.280)	0.306		
*stx*2a *stx*2c	19 (39.6)	29 (60.4)	1.916 (1.047–3.509)	0.032	1.878 (1.006–3.505)	0.048
*stx*2c	5 (17.9)	23 (82.1)	0.595 (0.223–1.587)	0.295		
Isolate relatedness				0.027		0.158
Outbreak-related	91 (23.1)	303 (76.9)	0.688 (0.494–0.958)		0.780 (0.552–1.101)	
Sporadic	100 (30.4)	229 (69.6)	1.454 (1.043–2.026)		1.282 (0.908–1.811)	

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
