# Peer review of "Epidemiology of Shiga Toxin-Producing Escherichia coli O157 in the Province of Alberta, Canada, 2009–2016"

_toxins, 2019, doi:10.3390/toxins11100613_

Round 1

Reviewer 1 Report

Lane 19: „Escherichia coli“  in italic letters please

Lanes 54-60: please write all names of genes in italic letters.

Table 2: In the fourth block – Shiga-toxin genotype - the lanes are out of alignment. Please correct it.

Here you wrote Shiga-toxin with a hyphen, which you did not in the rest of your manuscript. Please harmonize your wording.

Lanes 128 and 129: Please reword the first part of this sentence.

Lane 169 Please reword  „highly cyotoxic toxin „

Lanes 161-172: Please mention promoter regulation of the different Shiga toxin encoding phages, which also could be an explanation of higher or lower morbidity of different Shiga-toxin gene combinations.

Lane 188: You wrote: „Our results also point towards an important host-related risk factors for….“ Please delete the „s“ in factors.

Lane 270: You wrote: „Shiga toxin subtyping was informed by previous local data….“ What do you mean? Please reword this sentence.

Lane 319: Reference number 5: You wrote „escherichia coli o157:h7“. Please correct it.

Author Response

We sincerely thank Reviewer 1 for their time and thoughtful comments/observations.  Please find below a point-by-point response.  Changes unrelated to MDPI-introduced formatting issues have been highlighted in yellow.

Lane 19: „Escherichia coli“  in italic letters please

This has now been corrected.

Lanes 54-60: please write all names of genes in italic letters.

This has now been corrected. 

Table 2: In the fourth block – Shiga-toxin genotype - the lanes are out of alignment. Please correct it.

Please note that the formatting of tables has been modified by MDPI after submission by the authors.  We have taken the liberty to revert both tables to their original formatting, in order to facilitate peer review.

Here you wrote Shiga-toxin with a hyphen, which you did not in the rest of your manuscript. Please harmonize your wording.

This has now been corrected.

Lanes 128 and 129: Please reword the first part of this sentence.

The sentence has now been reworded.

Lane 169 Please reword  „highly cyotoxic toxin „

The sentence has now been reworded.

Lanes 161-172: Please mention promoter regulation of the different Shiga toxin encoding phages, which also could be an explanation of higher or lower morbidity of different Shiga-toxin gene combinations.

A new paragraph has been added to mention the stx promoter regulation by the different anti-terminator q genes.

Lane 188: You wrote: „Our results also point towards an important host-related risk factors for….“ Please delete the „s“ in factors.

This has now been corrected.

Lane 270: You wrote: „Shiga toxin subtyping was informed by previous local data….“ What do you mean? Please reword this sentence.

The sentence has now been reworded.

Lane 319: Reference number 5: You wrote „escherichia coli o157:h7“. Please correct it.

This has now been corrected. 

Reviewer 2 Report

The manuscript entitled “Epidemiology of Shiga Toxin-Producing Escherichia 2 coli O157 in the Province of Alberta, Canada, 2009-3 2016” provides very important and new information on the relationship between EHEC O157:H7 pathogenicity and development of human disease with some suggestions about a prediction of infection outcome and severity of disease.

However, several questions need to be addressed prior to the publication.

The authors retrospectively reviewed the STEC O157 cases in the province of Alberta over an 8-year period, 44 between years 2009 and 2016.

What about other non-O157 cases? Any cases of LEE-negative EHEC outbreaks? Any epidemiologic differences between O157 and the rest of cases?

Lines 78- 85: The authors report that infection activity was documented throughout the year, peaking during the summer months 78 (July – September) (56.2%). However, HUS peaked between January - March (n=5/39, 12.8%) and April - June 84 (n=12/39, 30.8%). So no HUS cases between July –September? What cause these differences? Any relation to Stx production/ genotype/clade?

The authors analyzed 191 cases of hospitalization, but only 118 cases of infection. How is this possible?

Line 134: In Discussion, the authors present a possible link between antibiotic treatment and HUS development, specifically in children. This part needs more attention with better in depth analysis. The data should be in Results section. Specifically, it is very important to compare the link between antibiotic use and HUS in children vs. adults (60 and older).

Author Response

The manuscript entitled “Epidemiology of Shiga Toxin-Producing Escherichia coli O157 in the Province of Alberta, Canada, 2009-2016” provides very important and new information on the relationship between EHEC O157:H7 pathogenicity and development of human disease with some suggestions about a prediction of infection outcome and severity of disease.

We sincerely thank Reviewer 2 for their time, thoughtful comments/observations, and acknowledgement of the importance of the present work.  Please find below a point-by-point response.  Changes unrelated to MDPI-introduced formatting issues have been highlighted in yellow.

However, several questions need to be addressed prior to the publication.

The authors retrospectively reviewed the STEC O157 cases in the province of Alberta over an 8-year period, between years 2009 and 2016.

What about other non-O157 cases? Any cases of LEE-negative EHEC outbreaks? Any epidemiologic differences between O157 and the rest of cases?

Laboratory procedures followed at the time significantly limited the capacity to detect non-O157 STEC isolates during the period studied (i.e. use of sorbitol-MacConkey agar and CHROMagar O157).   For that reasons we excluded non-O157 cases as their epidemiology would not be adequately represented and comparisons to O157 cases would not allow for meaningful conclusions.  Please see lines 274-277. 

Implementation of CHROMagar STEC has now been completed across the province and future iterations of STEC epidemiological work will be able to offer an accurate picture of also non-O157 infections, and compare those to O157 infections.  It will also allow for detection of non-O157 LEE-negative outbreaks, which have not been documented in Alberta to date.

Lines 78- 85: The authors report that infection activity was documented throughout the year, peaking during the summer months 78 (July – September) (56.2%). However, HUS peaked between January - March (n=5/39, 12.8%) and April - June 84 (n=12/39, 30.8%). So no HUS cases between July –September? What cause these differences? Any relation to Stx production/ genotype/clade?

Infection, related hospitalization and HUS all peaked during July-September.  This has been now clarified in the text.    

We believe that our dataset is underpowered to explore associations between seasonality and Shiga toxin genotypes.  This is now reflected in 3. Discussion. 

The authors analyzed 191 cases of hospitalization, but only 118 cases of infection. How is this possible?

In total, 762 cases of infection were analyzed.  Of these, 191 had STEC-related hospitalization, and 39 STEC-related HUS.  This has now been clarified in the text.

Line 134: In Discussion, the authors present a possible link between antibiotic treatment and HUS development, specifically in children. This part needs more attention with better in depth analysis. The data should be in Results section. Specifically, it is very important to compare the link between antibiotic use and HUS in children vs. adults (60 and older).

Access to provincial databases recording antibiotic use for inpatients and outpatients and linkage to the laboratory-based databases utilized was not feasible within the timelines for conclusion of the present work.  We agree and have acknowledged in 3. Discussion that future iterations of epidemiological studies in STEC would greatly benefit from such data linkage.

Round 2

Reviewer 2 Report

Accept in present form.